# The Role of LSD1 and LSD2 in Cancers of the Gastrointestinal System: An Update

**DOI:** 10.3390/biom12030462

**Published:** 2022-03-17

**Authors:** Gianluca Malagraba, Mahdieh Yarmohammadi, Aadil Javed, Carles Barceló, Teresa Rubio-Tomás

**Affiliations:** 1Translational Pancreatic Cancer Oncogenesis Group, Health Research Institute of the Balearic Islands (IdISBA), 07120 Palma de Mallorca, Spain; gianlucamalagraba@gmail.com; 2Central Tehran Branch, Department of Biology, Faculty of Sciences, Islamic Azad University, Tehran 1955847881, Iran; mahdiyeh.yarmohammadi@gmail.com; 3Cancer Biology Laboratory, Department of Bioengineering, Faculty of Engineering, Ege University, Izmir 35040, Turkey; adiljaved313@hotmail.com; 4Institut d’Investigacions Biomèdiques August Pi i Sunyer (IDIBAPS), 08036 Barcelona, Spain; 5School of Medicine, University of Crete, 70013 Herakleion, Crete, Greece

**Keywords:** demethylation, LSD1, LSD2, gastric cancer, liver cancer, pancreatic cancer, colorectal cancer

## Abstract

Epigenetic mechanisms are known to play a key role in cancer progression. Specifically, histone methylation involves reversible post-translational modification of histones that govern chromatin structure remodelling, genomic imprinting, gene expression, DNA damage repair, and meiotic crossover recombination, among other chromatin-based activities. Demethylases are enzymes that catalyse the demethylation of their substrate using a flavin adenine dinucleotide-dependent amine oxidation process. Lysine-specific demethylase 1 (LSD1) and its homolog, lysine-specific demethylase 2 (LSD2), are overexpressed in a variety of human cancer types and, thus, regulate tumour progression. In this review, we focus on the literature from the last 5 years concerning the role of LSD1 and LSD2 in the main gastrointestinal cancers (i.e., gastric cancer, liver cancer, pancreatic cancer, and colorectal cancer).

## 1. Introduction

According to the World Health Organization (WHO), cancer is a group of diseases that can develop in almost any organ or tissue when a group of abnormal cells grows uncontrollably beyond its normal limits, and can then invade other tissues (metastasis). In 2020, there were more than 19 million new cancer cases and almost 10 million deaths due to cancer, making this disease the second cause of death worldwide. Furthermore, up to 30% of these deaths are due to gastrointestinal cancers [1,2].

Understanding the mechanisms of cancer is key to developing efficient and specific treatments. Acquisition of cancer hallmarks largely depends on alterations in the genomes of neoplastic cells, including genome mutations, as well as epigenetic mechanisms which affect gene expression [3,4].

Epigenetics is commonly defined as the study of heritable changes in gene expression or chromosomal stability by DNA methylation, histone covalent modification (methylation, acetylation, ubiquitination…), or non-coding RNAs without a change in DNA sequence. Epigenetics plays a central role in cancer by altering proto-oncogenes and tumour suppressor transcription [5]. For a long time, methylation marks in histones were thought to be irreversible until the discovery of histone demethylases [6]. There are two families of histone demethylases: the larger Jumonji domain family and the smaller flavin-dependent lysine-specific demethylase (LSD) family formed by lysine-specific demethylase 1 (LSD1) and lysine-specific demethylase 2 (LSD2) [7]. 

LSD1 (also referred to as KDM1A/BHC110/AOF2) was the first human histone demethylase identified (2004). The LSD1 homolog, LSD2 (also referred to as KDM1B/AOF1), was identified the same year through a domain homology search of genomic databases and became the second human histone demethylase identified [6,7,8]. Both enzymes are characterized by the presence of an amine oxidase-like domain and a Swi3p, Rsc8p, and Moira (SWIRM) domain, which are unique to chromatin-associated proteins. Other than these two domains, LSD1 and LSD2 exhibit different structural architectures facilitating their association with different protein complexes and different genomic loci [7].

There are three structural domains present in LSD1 that are well conserved including the c-terminal amino oxidase-like (AOL) domain, the SWI3/RSC8/MOIRA (SWIRM) domain, and the flexible n-terminal region. The catalytic region of LSD1 resides on the AOL domain, which contains two lobes where one lobe connects with SWIRM, which further contains the follistatin domain (FSD)-binding site carrying oxidation, and the second lobe functions as a substrate recognition site. Therefore, the lobes form the catalytic centre displaying demethylation activity in the cavity. AOL domain also protrudes a Tower domain accompanying alpha-helices, which forms an interaction site with repressor element 1 (RE1) silencing transcription factor (REST) corepressor (CoREST) complex and is critical for H3K4 demethylase activity of LSD1. The extra-nucleosomal DNA can bind with the AOL domain along with the CoREST complex. The nuclear localization of LSD1 depends on the flexible n-terminal region of LSD1, which is not responsible for its demethylase activity. LSD1 has a specialized SWIRM domain incapable of binding to DNA and acts as an interaction site of LSD1 with its interacting partners [6,7,8].

LSD2 also referred to as AOF1 and KDM1B is a homolog of LSD1 possessing FAD-dependent amino-oxidase activity with no specificity towards non-histone substrates and specifically demethylates H3K4me1/2 along with H3K9me2 in the regions of the promoter genes associated with NF-kB proteins. The protruding TOWER domain is absent in LSD2 and it displays both AOL and SWIRM domains. Both LSD1 and LSD1 are known to exhibit FAD-demethylation activity, however, these proteins have other functions in cells including gene enhancer, promoter-binding, transcriptional repression, and activation properties. LSD2 changes the methylation dynamics of key transcriptional proteins such as NSD3, Cyclin T1, and Poly II and is known to interact with these transcriptionally activated genes via their coding regions as it assists in the regulation of the elongation process of transcription [6,7,8,9].

LSD1 specifically demethylates mono- and di-methylated histone 3 lysine 4 (H3K4me1/2), a histone mark linked to active transcription, and mono- and di-methylated histone 3 lysine 9 (H3K9me1/2), a histone mark linked to inactive transcription. Moreover, LSD1 forms a stable complex with corepressor CoREST. Although LSD2 presents less than 25% of sequence identity with LSD1 it has specificity for H3K4me1/2, unlike LSD1, LSD2 is not able to form a stable complex with CoREST. LSD2 catalyses the demethylation of their substrate using a flavin adenine dinucleotide (FAD)-dependent amine oxidation process [9]. Although LSD2 is found in most tumour cells, including tumours of the gastrointestinal system [10], it may also protect healthy adult cells from becoming malignant [11,12]. By joining with the RNA polymerase II (Pol II) elongating complex and demethylating H3K4me2, a crucial marker for epigenetic transcriptional activation, LSD2 functions as a core-pressor, facilitating transcription elongation [13,14]. 

Histone methylation is a reversible post-translational modification of histones that governs chromatin structure remodelling, genomic imprinting, gene expression, DNA damage repair, and meiotic crossover recombination, among other chromatin-based activities [15]. Thus, LSD2 may be involved in transcriptional programmes distinct from those of LSD1 [6]. Moreover, new noncatalytic LSD interactions have been highlighted as being relevant in cancer [8].

In addition, both LSD1 and LSD2 target non-histone proteins. For example, LSD1 demethylates the well-known tumour suppressor protein p53 [16], as well as myosin phosphatase target subunit 1 (MYPT1) [17] and SRY (sex determining region Y)-box 2 (SOX2) [18] and these activities are thought to promote tumour progression.

Indeed, LSD1 is highly expressed in many cancer types including breast, prostate, oesophageal, bladder and lung cancer, and neuroblastoma and acute myeloid leukaemia [7]. Understanding of the biological role of LSD2 is not as broad as that of LSD1. However, it is known that LSD2 plays a role in oncogenesis, although its mechanisms are less clear [6].

In this review, we summarise the latest findings regarding LSD1 and LSD2 in cancers of the gastrointestinal system: gastric cancer (GC), liver cancer, pancreatic cancer, and colorectal cancer (CRC). 

## 2. Gastric Cancer

GC is the fifth most common cancer diagnosed worldwide and the third leading cause of death by cancer. The overall survival has improved due to early cancer detection and improved standard-of-care cancer therapies, including tailored treatment based on tumour molecular biology [19,20]. 

### 2.1. LSD1 in Gastric Cancer

LSD1 has been shown to promote gastric cancer proliferation [21]. More importantly, overexpression of LSD1 is involved in many pathological processes of gastric cancer, such as proliferation, apoptosis, and metastasis of various GC cells [22].

Recently, a number of studies have highlighted the critical roles played by long noncoding RNAs (lncRNAs) in the pathogenesis of several types of human cancer, especially in GC [23]. More importantly, lncRNAs can directly bind to LSD1 and may function as a scaffold, thereby repressing underlying Krüppel-like factor 2 (KLF2) target and large tumor suppressor kinase 2 (LATS2) expression [24]. LSD1 and lncRNAs have a role in carcinogenesis and cancer spread by suppressing tumour suppressors or activating oncogenes through various methods such as epigenetic alteration, alternative splicing, RNA decay, and posttranslational modification regulation [25]. Lysine (K)-specific demethylase 1A (LSD1) as the core of the LSD1/CoREST/REST repressor complex and histone demethylase could specifically demethylate H3K4me1/2. Recent studies have shown that multiple tumour-related lncRNAs regulated cancer progression through interactions with enhancer of Zeste 2 Polycomb repressive complex 2 subunit (EZH2) and LSD1 [21].

#### 2.1.1. LSD1 and LincRNAFEZF1-AS1

LSD1 can hyperactivate GC cells aided by LincRNAFEZF1-AS1 to repress p21 expression [26]. Shin et al. showed that endogenous FEZF1-AS1 was enriched in the anti-LSD1 RIP fraction in AGS and SGC-7901 gastric adenocarcinoma cell lines, concluding that LSD1 could promote GC cell proliferation [27]. Knocking down LSD1 by si-RNA in AGS and SGC-7901 cells upregulated p21 protein levels [28]. Moreover, p21 was enhanced in AGS and SGC-7901 cells treated with an LSD1 inhibitor compared to untreated cells, suggesting that FEZF1-AS1 regulates p21 through LSD1-mediated demethylation. To sum up, this study reported that LSD1 could directly bind to the promoter region of p21 and mediate H3K4me2 modification, while knockdown of FEZF1-AS1 led to reduced LSD1 and increased H3K4me2 demethylation ability [21]. 

#### 2.1.2. LSD1 and lncRNA HOXA11-AS

HOXA11-AS is a GC–specific upregulated lncRNA since its expression levels are increased in GC compared to normal gastric tissues [29]. Sun et al. investigated the potential mechanisms of HOXA11-AS and LSD1 in GC cells, concluding that HOXA11-AS RNAs are more prevalent in the nucleus of a panel of GC cells [30]. Additionally, it was predicted that lncRNA HOXA11-AS mediates EZH2 and LSD1 interaction [31]. At the same time, DNMT1, EZH2, and LSD1 are also likely to be recruited by other sequence-specific transcription factors [32]. Since HOXA11-AS potentially binds LSD1 in GC cells, it is predicted to function as a scaffold to regulate PRSS8 and KLF2 transcription [33].

#### 2.1.3. LSD1 and Long Noncoding RNA FOXD2-AS1

There are increasing data indicating that FOXD2-AS1 serves as an important modulator in biological processes and is dysregulated in GC, in which it could potentially serve as a tumour biomarker [34]. It has been shown to act as a tumour inducer in GC, in part through EphB3 inhibition through direct interaction with EZH2 and LSD1 [35]. FOXD2-AS1 accumulates in GC and is upregulated in GC cells and positively correlates with large tumour size, advanced pathological stage, and poor prognosis [35,36]. Gene set enrichment analysis in Gene Expression Omnibus (GEO) datasets revealed that cell cycle and DNA replication-associated genes were enriched in patients with high FOXD2-AS1 expression [37]. Loss of FOXD2-AS1 function inhibited cell growth through cell cycle inhibition in GC, while upregulation of FOXD2-AS1 expression promoted cancer progression [38,39]. 

#### 2.1.4. LSD1 and LINC00673

Long noncoding RNA LINC00673 is overexpressed in GC [40], in which it acts as a scaffold for LSD1 and EZH2 and represses KLF2 and LATS2 expression. Indeed, LINC00673 binds directly to EZH2, LSD1, DNMT1, and STAU1 in GC cells. When BGC823 and AGS cells were treated with both EZH2 and LSD1 siRNAs the expression of the tumour suppressor LATS2 and KLF2 was enhanced, whereas no effect was observed on CADM4 expression [24].

### 2.2. LSD2 in Gastric Cancer

ADPGK antisense RNA 1 (ADPGK-AS1) promotes GC development through upregulation of LSD2 by sponging miR-3196, making it a potential new prognostic biomarker and therapeutic target for GC patients, and ADPGK-AS1 is dramatically overexpressed in GC cell lines compared to normal gastric epithelial cell lines [11]. Furthermore, GC patients with high ADPGK-AS1 levels had a poorer overall survival rate than GC patients with low ADPGK-AS1 levels [41]. Inhibition of ADPGKAS1 markedly accelerated GC cell apoptosis [42] and downregulated LSD2 protein levels, and this phenotype was partially rescued by inhibition of miR3196 [43]. Similarly, upon LSD2 overexpression, cell proliferation due to inhibition of ADPGKAS1 was mostly restored and the facilitating effect of inhibition of ADPGKAS1 on apoptosis was partially abolished [43]. Increased expression of ADPGKAS1 and LSD2 may be directly related to the PI3K/AKT/mTOR signalling pathway in GC tissues [11]. In addition, ADPGKAS1 downregulates p53 through the regulation of LSD2 [44] 

There is little literature regarding the role of LSD1 and LSD2 in GC (Table 1), and the main conclusions are summarised in Figure 1.

## 3. Liver Cancer

Cancers of the liver pose a great challenge to global health with the overall cases expected to rise to more than one million by the year 2025 [45,46]. Liver cancers are mostly driven by hepatocellular carcinomas (HCC), which are associated with hepatitis B and C infections leading to cirrhosis and other chronic liver diseases [45]. The primary liver cancer is HCC with risk factors involving long-term liver diseases [47] and higher consumption of alcohol along with fat accumulation in the organ (liver) [48]. The standard diagnostic procedures for liver cancer include liver biopsy, imaging tests, and liver function tests of the blood. The treatment strategies vary along the spectrum of the disease in terms of severity and liver function of the patients. The general treatments for liver cancer including HCCs are surgical resection of cancer, transplant surgery to replace the diseased liver with a liver from a healthy donor, ablation therapy for killing cancer cells by microwaves or radiation, heat or cold (cryoablation), chemotherapy, radiation therapy, targeted drug delivery approaches, and immunotherapy [49,50,51,52,53]. 

### 3.1. LSD1 in Liver Cancer

The poor prognosis of HCC has been linked to the differentiation and self-renewal capacities of the cancer stem cells (CSCs), which are characterised by different cellular markers [54]. The self-renewal capacity of CSCs is a topic of interest in oncology and the mechanisms remain to be investigated; however, the tumorigenicity of these specialised cells has been associated with players involved in epigenetic dysregulation such as lysine demethylases including LSD1 that is a chromatin modification factor and acts to demethylase histone H3 lysine 9 (H3K9) along with the histone 3 lysine 4 (H3K4). These histones generally function in the regulation of stem cells and genome instability leading to cancers [55,56]. The hematopoietic and embryonic stem cells have pluripotency that is regulated from the epigenetic regulator LSD1 [57,58,59].

Since LSD1 is involved in the regulation of pluripotent cancer cells, the inhibition of this important class of enzymes poses an important treatment or targeting strategy for cancer therapy [60,61]. The development and stemness of HCC are promoted due to suppression of the negative regulators such as beta-catenin in Lgr5p cancer cells expressing higher levels of LSD1 [62]. The CSCs associated with HCC exhibit higher levels of LSD1 as compared to non-CSCs, and the higher expression of LSD1 is reduced in differentiated CSCs. The self-renewal capacity of non-CSCs is increased due to overexpression of LSD1 showing that LSD1 is involved in the tumorigenicity of HCC. The high activity of LSD1 is inversely correlated with acetylation as the level of acetylation of LSD1 is reduced in CSC, which shows higher self-renewal capability [62]. The activation of LSD1 results from the induction of sirtuin SIRT1 by Notch signalling, which is involved in the self-renewal of CSCs and represents one of the mechanisms by which LSD1 exerts its function in the promotion of HCC (Figure 2). CSC self-renewal mediated through Notch-3 signalling results from cancer-associated fibroblasts as upstream drivers with higher LSD1 expression. Therefore, the microenvironment of the tissue inside liver cancer plays a significant role in driving the role of LSD1 in Notch signalling and stemness of the CSCs, which can be targeted for therapeutic purposes [62].

The expression of LSD1 is higher in liver cancer tissues compared to noncancerous tissue adjacent to the tumour as determined by immunohistochemistry and Western blotting [63]. Furthermore, lower tumour stages also exhibit lower expression of LSD1 as compared to higher tumour stages. In liver cancer cell lines, the knockdown of LSD1 results in decreased proliferation along with reduced expression of cMyc and Bcl-2, implying that LSD1 is involved in the survival of cells [63]. Additionally, Kim et al. [64] estimated approximately 77% of the 303 patients with HCC (*n* = 303 cases of HCC) where positive for LSD1 protein expression by immunohistochemical analysis. Higher expression of LSD1 was associated with poorer outcomes for HCC, especially for disease-free survival and overall survival. The authors also used CRISPR/Cas9 system to knock out LSD1 in HCC cell lines and showed that LSD1 is involved in the control of growth rate. Furthermore, the LSD1 knockout resulted in increased H3K9me1/2 and HeK4me1/2 levels along with reduced S-phase population, probably by targeting retinoic acid pathway [64]. In summary, according to this study, LSD1 could be a potential therapeutic target for HCC and needs to be explored further for its role in HCC. An overview of LSD1 involvement in HCC is shown in Figure 2.

#### LSD1 Inhibition as a Treatment Strategy for Liver Cancer

Since overexpression of LSD1 has been associated with various forms of cancer and silencing LSD1 reduces the migration, invasion, and proliferation of cancer cells, inhibition of LSD1 is considered one of the clinical interventions in liver cancer [62,63,65,66]. Therefore, LSD1 inhibitors are used as potential anti-cancer drugs [67]. Drug resistance in cancer is one of the major problems that requires attention; for example, oxaliplatin is a drug used as a chemotherapeutic agent in the therapy of HCC. However, drug resistance to oxaliplatin is a challenge for healthcare professionals and currently jeopardizes its use as a therapy for HCC. One of the mechanisms by which HCC attains drug resistance is via up-regulation of long non-coding RNAs such as LINC01134. The LINC01134 promoter is demethylated by LSD1 and leads to up-regulation of LINC01134, which, in turn, stabilizes p62 and assists in the activation of anti-oxidative stress pathway in HCC tissues and cells [68]. Therefore, LSD1 inhibition by specific inhibitors can be a potential option against chemo-resistance in HCC. Recently, novel small molecule inhibitors against LSD1, such as coumarin analogues and benzofuran derivatives, have been synthesized and can be utilized for targeting the cellular activity of LSD1 [69,70]. Moreover, tertiary sulphonamide derivatives exhibiting dual properties of inhibition of tubulin polymerization and LSD1 inhibition have recently been implicated as potential treatment for liver cancer [71].

### 3.2. LSD2 in Liver Cancer

In the context of liver cancer, the Huh7 and Hep3b HCC cell lines exhibit upregulated LSD2 according to a comprehensive study designed to identify the targets involved in epigenetic alterations of these cells [72]. In another study involving HCC cell lines for sorafenib resistance, the expression of LSD2 did not change and LSD1 was found to be a regulator of drug resistance in which LSD2 depletion resulted in no apparent change in drug sensitivity [73]. Apart from these aforementioned studies, there is a dearth of data on LSD2 in the context of liver cancer and it is a potential subject of future research which needs to be explored.

The recent publications regarding the role of LSD1 and LSD2 in liver cancer are summarised in Table 2.

## 4. Pancreatic Cancer

LSD1 seems to be involved in the progression of pancreatic cancer as it is a target for various long non-coding RNAs involved in this process [22]. Lian et al. [74] identified that HOXA cluster antisense RNA 2 (HOXA-AS2) exerts an oncogene function via interaction with LSD1. HOXA-AS2 is upregulated in pancreatic cancer tissue, promoting pancreatic cancer cell proliferation and reducing apoptotic rates in vitro (PANC-1 and BxPC-3 cell lines), and plays an important role in pancreatic cancer cells tumorigenesis in vivo (xenograft assays using BxPC-3 cells into BALB/c mice). Using RNA-protein interaction prediction, it was found that HOXA-AS2 binds with LSD1 and EZH2. RNA immunoprecipitation assays performed using LSD1 antibodies prove that HOXA-AS2 does, in fact, bind to LSD1 in BxPC-3 cells. Moreover, a positive correlation between LSD1 mRNA levels and HOXA-AS2 expression has been observed thanks to data on pancreatic cancer gene expression (GSE15471) obtained from the Gene Expression Omnibus database (GEO). Altogether, these results suggest that LSD1 functions as an oncogene in pancreatic cancer cells as it is involved in a lncRNA-HOXA-AS2/EZH2/LSD1 complex which promotes cell proliferation [74]. 

HOXA-AS2 is not the only lncRNA that has been associated with LSD1 in pancreatic cancer cells. Double homeobox A pseudogene 10 (DUXAP10)-derived lncRNA also plays an important role in pancreatic cancer cells. This lncRNA is upregulated in human pancreatic cancer tissues and is associated with poor prognosis/advanced tumour-node-metastasis (TNM) stages. According to Lian et al., DUXAP10 positively correlates with cell proliferation pathways, reduces apoptotic rates, enables higher cell migration and invasion rates in vitro (PANC-1 and BxPC-3 cell lines) and also inhibits pancreatic cancer cell tumorigenesis in vivo (xenograft assays using BxPC-3 into BALB/c mice). Since lncRNAs are known to interact with RNA binding proteins, the authors performed an RNA protein interaction prediction assay, concluding that DUXAP10 interacts with LSD1 and EZH2. This interaction was verified via RNA immunoprecipitation assays performed using LSD1 antibodies in BxPC-3 cells. Furthermore, a positive correlation was found between DUXAP10 expression and LSD1 mRNA levels. The reduction in BxPC-3 cell viability when treated with si-DUXAP10 and si-LSD1 is greater than when treated only with si-LSD1. Thus, LSD1 seems to play an important role in pancreatic cancer cell proliferation as it binds to DUXAP10 [75]. 

Regarding LSD2, Wang et al. proposed that it may be involved in the development of pancreatic cancer. An increased LSD2 expression in pancreatic cancer cells has been established by immunohistochemistry using an antibody against LSD2 in cancer tissue samples compared with their respective matching paracancerous tissue samples, as well as by immunoblotting using human pancreatic cancer cell lines (BxPC-3, CFPAC-1, PANC-, SW1990) and a normal human pancreatic duct epithelial cell line (HDPE6-C7). This increase in LSD2 expression is accompanied by suppression of cell proliferation and increase in apoptosis. When the pancreatic cancer cell lines PANC-1 and SW1990 are treated with shLSD2 resulting in LSD2 knockdown cells, in vitro cell growth is significantly reduced and apoptosis increases. Moreover, caspase 3 and caspase 7 levels and activity are increased in PANC-1 and SW1990 knockdown cells. Thus, LSD2 expression is an important determinant of apoptosis in these cell lines [76].

The main findings concerning the role of LSD1 and LSD2 in pancreatic cancer are summarised in Table 3 and Figure 3.

## 5. Colorectal Cancer

### 5.1. LSD1 in Colorectal Cancer

LSD1 has been reported to be overexpressed in CRC. Overexpression of LSD1 in CRC, as in various other cancer types, facilitates proliferation, migration, invasion and stemness, and is associated with higher TNM stages [77,78,79]. Despite these findings being coherent with other studies about LSD1 in cancer, other studies suggest that LSD1 is negatively associated with CRC tumorigenesis. Ramírez-Ramírez et al. established that loss of LSD1 is significantly associated with metastasis to lymph nodes and TNM stages III-IV after analysing CRC tissues. The data presented in this study suggest that LSD1 gene suppression is key during metastasis. These opposite facts may be due to the genetic background of the cells as well as the effect of environmental signals on LSD1. Additionally, consider that metastasis is a different process from carcinogenesis itself [80]. Furthermore, Carvalho et al. established that those patients who display low LSD1 expression levels (analysed by immunohistochemistry) in their CRC tumours experienced significantly lower disease-specific survival and disease-free survival. In this case, these results may differ from other studies with limitations such as low sample size, different methods to assess LSD1 expression and unknown factors in the genetic and/or environment background which may affect LSD1 regulation [81].

As LSD1 has been associated with invasion and metastasis in some cancer types, the association between LSD1 and stemness features in CRC has been studied by analysing CD133+ CRC cells. CD133 is a surface marker for CRC stem cells associated with higher viability and colony formation rate. LSD1 expression is higher in CD133+ CRC cells and is associated with stemness, not only in vitro, but in vivo as well (xenograft assays using SW620 in BALB/c mice). LSD1 knockdown increases the apoptotic rate and further decreases cell viability, colony formation rate, migration, and invasion of CD133+ cells in response to anti-cancer drugs [77]. 

To better understand the role of LSD1 in CRC, Chen et al. performed an LSD1 downstream target analysis in which they found 4 key LSD1-target genes associated with proliferation, metastasis, and invasion: CABYR, FOX2, TLE4 and CDH1. Moreover, they established that LSD1 overexpression mediates CABYR and CDH1 downregulation by decreasing the levels of H3K4me1 and H3K4me2 at their promoter regions [78]. 

Some studies associate LSD1 with PI3K/AKT, thus partly explaining LSD1-mediated cell proliferation. LSD1 correlates with AKT phosphorylation (pS473-AKT). LSD1 regulation effects on pS473-AKT do not rely on LSD1 catalytic activity, but rather on its scaffolding function for the CoREST complex. By regulating AKT, LSD1 regulates Snail protein stability, an epidermal-mesenchymal promoting transcription factor, at least in PIK3CA mutated cells in which LSD1 is highly expressed [82]. LSD1 in CRC also affects PI3K/AKT via RIOK1. Furthermore, LSD-dependent demethylation of RIOK1 in CRC significantly stabilizes RIOK1 proteins, promoting CRC cell proliferation and migration through PI3K/AKT [83].

LSD1 not only affects the proliferation process of CRC cells, but also the differentiation process. In BRAF mutant CRC, LSD1 is required for maintenance of enteroendocrine progenitors. Thus, LSD1 is associated with a secretory phenotype in BRAF mutant CRC. Not only is LSD1 associated with the differentiation of enteroendocrine cells, but it is also associated with TFF3 expression and protein secretion, which is critical for cell survival during growth factor signalling; in fact, it is involved in AKT phosphorylation in S437 [79].

In terms of LSD1 regulation, after screening an 80-gene deubiquitinase bank, the USP38 protein was identified as a LSD1 deubiquitinase stabilizing its protein levels via posttranslational modification. Therefore, by removing the ubiquitin chain from the LSD1 protein, USP38 enhances the activity of the signalling pathways activated by LSD1, enhancing proliferation, colony formation and antiapoptotic proteins [84].

Other relevant proteins have been positively correlated with LSD1, such as Tenascin C (TNC), and Tetraspanin 8 (TSPAN8), among others [85,86]. In fact, data show that TSPAN8, a transmembrane protein, enhances the tumorigenicity of CRC as it produces the same effects as LSD1. It has been established that LSD1 upregulates TSPAN8 reducing H3K9me2 occupancy on its promoter. Considering that TSAPN8 and LSD1 depletion results in an upregulation of E-cadherin and ZO-1, and a downregulation of n-cadherin, Vimentin, Slug and Snail, TSPAN8 may promote epithelial-mesenchymal transition in an LSD-1 dependent manner [86].

Finally, in addition to the aforementioned pathways, LSD1 plays a significant role in lipid regulation in CRC. As in vitro data suggest, LSD1 may play a critical role in cancer lipid metabolism. Treatment with an LSD1 inhibitor produces lipid dysregulation specially in sphingolipids and glycolipids, enhancing lipids such as ceramides and sphingomyelins. These lipids are bioactive and are involved in signalling pathways including apoptosis. Therefore, these findings may establish a connection between LSD1-mediated apoptosis and lipid metabolism [87]. 

#### LSD1-Associated Noncoding RNAs Involved in Colorectal Cancer Progression

Several long noncoding RNAs (lncRNAs) have been associated with LSD1 in CRC. Pseudogene-expressed lncRNA DUXAP10 is upregulated in human CRC and positively correlates with tumour size, TNM stage and lymph node metastasis. Thanks to RNA-protein interaction prediction and posterior ChIP assays, it has been established that DUXAP10 binds to LSD1 in some CRC cell lines mediating H3K4me2 demethylation in the p21 and PTEN promoter regions, which explains the DUXAP10 proliferation effects [88]. Similarly, lncRNA DUXAP8 is upregulated in human CRC cells and accelerates their proliferation via binding to LSD1 and EZH2 [89]. In addition to the aforementioned lncRNAs, ZEB2-AS1 and HOXA-AS2 are upregulated in CRC and are associated with increased proliferation rate, tumour size, higher TNM stage and lymph node metastasis. Both ZEB-AS1 and HOXA-AS2 bind to LSD1 possibly accelerating the cell proliferative rate [90,91]. Regarding HOXA-AS2, data suggest that it binds to LSD1 and silences p21 and KLF2 transcription as it enhances H3K4me2 demethylation in their promoters [91]. 

MicroRNAs are other important noncoding RNAs in cancer. miR-137-3p is one of the microRNAs that are downregulated in various cancers, including CRC. Its downregulation is associated with invasiveness of CRC cells. This microRNA negatively regulates LSD1 expression, thereby decreasing cell adhesion and migration. miR-137-3p and LSD1 both respond to hypoxia and data suggest that a hypoxic CRC environment could induce mi-R-137-3p repression, thus derepressing LSD1 expression to induce the EMT program and tumour metastasis [92]. 

### 5.2. LSD2 in Colorectal Cancer

Similar to LSD1, LSD2 is upregulated in human CRC cells, both in vitro and in vivo. High levels of LSD2 promote cell proliferation, DNA synthesis, colony formation rate and colony size. Data show that LSD2 induces cell cycle progression and reduces apoptosis by reducing apoptosis inhibitor Bcl-2 levels and increasing cleaved caspase 3, cleaved caspase 9 and apoptosis sensor BAX levels. Moreover, LSD2 downregulates p53 expression through H3K4me2 demethylation in the p53 promoter region, thus driving the cell cycle through p53-p21-Rb. In fact, LSD2 is associated with a decrease in p53 and p21 protein levels and an increase of Rb protein levels [44]. 

The main findings concerning the role of LSD1 and LSD2 in CRC are summarised in Table 4 and Figure 4.

## 6. Therapeutic Implications of LSD1 and LSD2 in Various Cancers

The majority of research defining the role of LSD1 in metabolism is connected to cancer biology, and little is known about the participation of LSD1 in endothelial cell proliferation, which is critical for circulatory functions and cancer development and metastasis [93]. The expression of LSD1 being higher in cancers leads to a proposition of it being an anti-cancer target and various studies report that the pharmacological inhibition of LSD1 could be a potential treatment strategy for various cancers [94]. The synthetic compounds, peptides, and natural products have been reported as numerous LSD1 inhibitors [95,96]. New irreversible LSD1 inhibitors have been designed by using tranylcypromine (TCP) scaffolds and have been tested in clinical trials together with other agents. In addition, other LSD1-specific inhibitors such as ORY-2001, IMG-7289, GSK2879552, and ORY-1001, can target LSD1 with more potency, compared to TCP and they may be useful for other cancer type treatments, such as acute myeloid leukaemia [67]. Nevertheless, to date, there are no LSD1 or LSD2 inhibitors approved for neither pancreatic cancer nor colorectal cancer [97,98]. Fang et al. summarized the toxicity profiles of various LSD1 inhibitors used on patients in clinical trials and have provided recommendations for designing the dosing regimens along with the in-depth mechanistic studies required for LSD1 inhibitors in pre-clinical setups and the development of safer profiles based on potent and selective inhibitors [67]. GSK2879552, which is an LSD1-specific inhibitor, showed lesser potency for cell proliferation assays as compared to dual inhibitors that target HDAC1/2 in vitro [99]. Therefore, the dual inhibitors targeting LSD1 and HDAC may confer better outcomes in clinical applications for the treatment of cancers. Moreover, apart from demethylase activity exhibited by LSD1, there are other functions of LSD1 such as interaction and degradation of a tumour suppressor FBXW7 [100]. Therefore, inhibitors targeting the non-enzymatic activity of LSD1 can also be considered as potential anti-cancer therapeutic options. Remarkably, some of the LSD1 inhibitors might also have an effect on the activity of LSD2 because of the similarities in their catalytic domains, and LSD2-specific inhibitors are, up to now, unavailable [10].

LSD2 is also a promising epigenetic target for treating various types of cancer. However, despite significant advances in understanding the structural and functional aspects of LSD2 function, the underlying molecular mechanisms involved in cancer biology have not been conclusively established. LSD2 was initially identified to maintain low levels of H3K4 methylation in the renal region. This enzyme can promote gene expression by removing the inhibitory H3K4 methylation mark (H3K4me1 or H3K4me2) [101]. Recent reports have shown that aberrant LSD2 expression is often associated with dysregulation of histone activity, which contributes to aberrant gene expression in some cancers such as glioblastoma, breast cancer, pancreatic cancer, and GC [102].

As described previously, LSD1 is overexpressed in various cancer cells including ER-negative breast cancer, bladder, prostate, lymphoid neoplasm, lung cancer, and acute myeloid leukaemia cells, and is associated with poor prognosis of various tumours. LSD1 promoting AML via H3K4 demethylation for the regulation of erythrocyte differentiation along with mediating GFi1 and GF1ib for regulating hematopoietic differentiation are the mechanisms that can be targeted for AML differentiation [103,104]. Moreover, LSD1 has been linked with the pathways involved in promoting growth, metastasis, invasion, migration, and proliferation of cancer cells [67,105]. Therefore, LSD1 presents a viable target as a therapeutic target for various cancers. For example, the SIN3A/HDAC complex involved in the oncogenic potential and survival of breast cancer cells also exhibits LSD1 as one of the functional components [106]. In neuroblastoma, LSD1 displayed an inverse correlation with differentiation and is upregulated in poorly differentiated cancer cells [107]. LSD1 is reported to be an initiating factor of epithelial-mesenchymal-transition (EMT) and is overexpressed in bladder cancer cells [108]. LSD1 also binds with one of its interacting partners ZNF217 and activates key gene networks responsible for prostate cancer survival and proliferation. The demethylase independent functions can be blocked in prostate cancer by targeting LSD1 using its inhibitor named SP-2509 [109]. LSD1 is also overexpressed in Small Cell Lung Cancer (SCLC) where suppression of LSD1 leads to modulation of NOTCH-ASCL1 axis causing inhibition of tumorigenesis and chemoresistance. LSD1 inhibitor ORY-1001 showed significant therapeutic effects in PDX models [110]. Immunotherapy for increasing anti-tumour effects has also been considered as a potential route of application for LSD1 inhibition as LSD1 expression is reported as inversely correlated with cytotoxic T cell attracting chemokines along with the programmed death-ligand 1 (PD-L1) in triple-negative breast cancer (TNBC). Therefore, LSD1 suppression can be effective in providing a tool for adjuvant treatment options in combination with immunotherapy [111]. The poorly immunogenic tumours of the gastrointestinal system can also be explored for the role of LSD1 inhibition as a novel management strategy.

Growing evidence from these past years shows that LSD1 and LSD2 are key proteins in enabling cancer cells to achieve cancer hallmarks such as genome instability, sustaining proliferative signals or activating invasion and metastasis, not only in cancers of the gastrointestinal system, but also in cancers such as prostate cancer, neuroblastoma, lung cancer and acute myeloid leukaemia [67]. Therefore, it is not bizarre that these discoveries attract the attention of researchers in whether treatments targeting LSD1 and/or LSD2 are feasible. Compounds are being tested in pre-clinical studies obtaining a diverse range of results. Although studies knocking down/out LSD1 show promising results, it is important to bear in mind that results may differ from studies using pharmacological enzymatic inhibition as it is known that noncatalytic LSD1 role is also important in cancer development [112]. The same is applicable for LSD2 for which inhibition has shown also promising results in vitro in some cancers like breast cancer [113], prostate cancer [114], or gastrointestinal system cancers. 

Diverse pharmacological LSD1 inhibitors have been developed, the first one being tranylcypromine (TCP). Nowadays, there are many LSD1 inhibitors with more specificity and potency than TCP [67]. Until 2021, several LSD1 inhibitors including irreversible inhibitors tranylcypromine (TCP), ORY-2001, ORY-1001, GSK-2879552, INCB059872, IMG-7289, TAK-418, and reversible inhibitors CC-90011 and SP-2577 have entered clinical evaluation as mono- or combined therapy for diseases, particularly for AML and small lung cancer cells (SCLC). LSD1 inhibition has aroused interest, especially in acute myeloid leukaemia (AML) and small cell lung carcinoma (SCLC), in fact, TCP derivatives started being studied in oncology clinical trials to assess their value as AML and SCLC treatment [7]. Results for these studies are promising. The novel LSD1 inhibitors and combination regimens provide new therapeutic strategies for AML treatment. In fact, combination therapy may help overcome LSD1 inhibitor resistance. The fact that LSD1 and LSD2 may act synergistically with other pathways opens the door to exploiting the use of LSD1/2 inhibitors in combination therapies [112]. For example, it is shown that LSD1 inhibition is highly synergistic with mTOR inhibitors in endometrial cancer, increasing the limited success that mTOR inhibitors presented [115].

## 7. Conclusions

LSD1 was the first of several protein lysine demethylases discovered. This enzyme is in charge of demethylating mono- and dimethylated lysines in histone H3 at positions 4 or 9. (H3K4 and H3K9, respectively). The extensive ramifications of LSD1-controlled demethylation explain the substrate specificity of histones harbouring lysine-specific demethylase and many crucial non-histone proteins, especially transcription factors, chromatin-regulating proteins, and also tumour suppressor proteins [8]. The control of histone protein demethylation by LSD1 is a key component of the regulatory mechanism of carcinogenesis. As data shows how LSD1 and LSD2 play an important role in cancer development, it is not rare to find a growing interest in LSD inhibition as an anticancer therapy. Since its discovery in 2004, various attempts have been made to uncover the function of LSD1 in different contexts including viral infections and cancers. Further research is needed to elucidate the role of LSD1 and LSD2 in the pathogenesis of cancers of the gastrointestinal system. We expect that further characterisation of the molecular function of LSD1 and LSD2 will pave the way for the development of novel and innovative therapeutic strategies for different types of cancer, including GC, liver cancer, pancreatic cancer, and CRC.

## Figures and Tables

**Figure 1 biomolecules-12-00462-f001:**
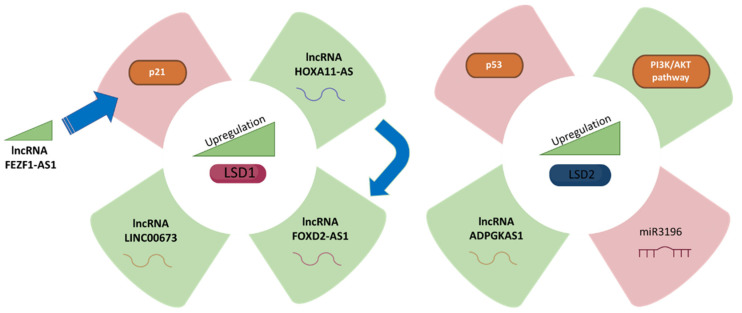
LSD1 (**left** panel) and LSD2 (**right** panel) expression and its correlation with gastric cancer fate. The green colour represents upregulation while purple represents downregulation of different molecules, which are related to LSD1 and LSD2 upregulation, respectively.

**Figure 2 biomolecules-12-00462-f002:**
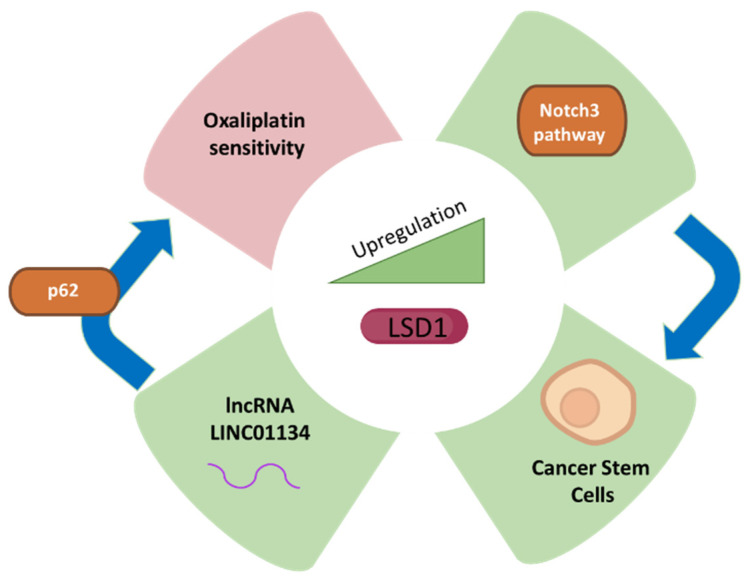
LSD1 expression and its correlation with liver cancer fate. The green colour represents upregulation or increase while purple represents decrease, which are related to LSD1 upregulation.

**Figure 3 biomolecules-12-00462-f003:**
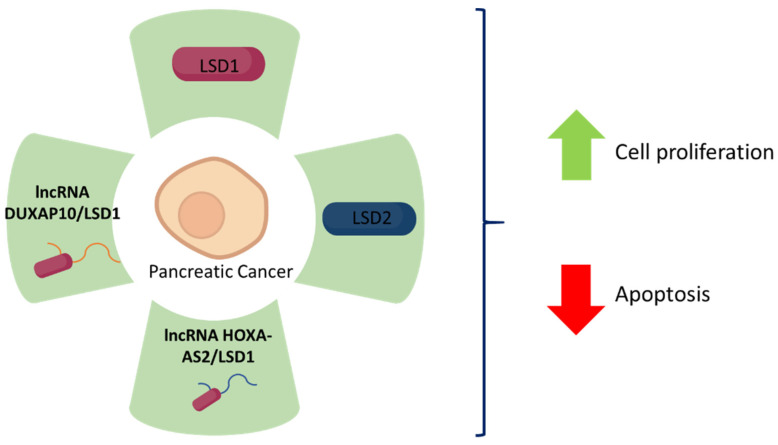
LSD1 and LSD2 expression and its correlation with pancreatic cancer fate. The green colour represents upregulation of each specific molecule, which all together leads to increased pancreatic cancer cell proliferation and decreased apoptosis.

**Figure 4 biomolecules-12-00462-f004:**
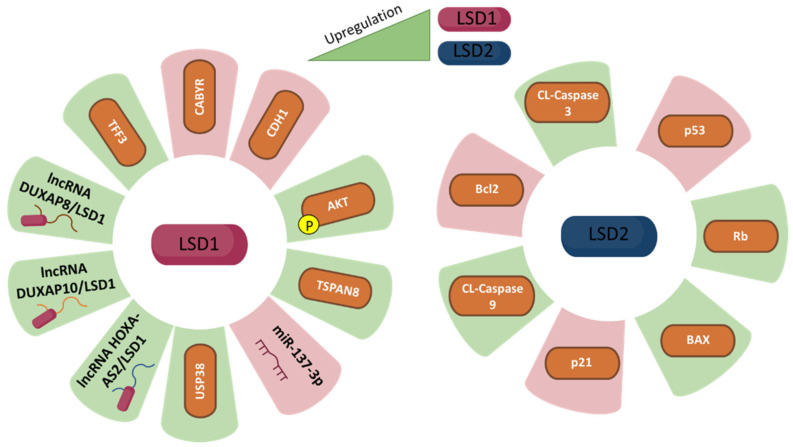
LSD1 (**left** panel) and LSD2 (**right** panel) expression and its correlation with colorectal cancer fate. The green colour represents upregulation while purple represents downregulationof each specific molecule, which are related to LSD1 and LSD2 upregulation, respectively.

**Table 1 biomolecules-12-00462-t001:** Recent studies on LSD1/2 in gastric cancer.

LSD	Model	Main Findings	Reference
LSD1	The GC cell lines (MGC-803, SGC-7901 and AGS)	LSD1 hyperactivates gastric cancer cells with the assistance of LincRNAFEZF1-AS1 to repress p21 expression. LSD1 promotes GC cell proliferation.	[21]
LSD1	(BGC823, SGC7901, MGC803, AGS) and normal gastric mucosa GES1 cell lines.	HOXA11-AS potentially binds LSD1 in GC cells and functions as a scaffold to regulate PRSS8 and KLF2 at transcriptional level.	[30]
LSD1	The GC cell lines (GES1, BGC823, AGS, MKN45, SGC7901 and MGC803)	FOXD2-AS1 is dysregulated GC. FOXD2-AS1 promotes GC tumorigenesis partly through EZH2- and LSD1-mediated EphB3 downregulation.	[35]
LSD1	The GC cell lines (GES1, BGC823, AGS, SGC7901 and MGC803)	LINC00673 works as a scaffold for LSD1 in GC. LINC00673 binds directly to LSD1 in GC cells. LINC00673 and LSD1 are involved in regulating CADM4, LATS2, and KLF2.	[24]
LSD2	The GC cell lines (BGC-823, MGC-803, AGS and SGC-7901)	Cell proliferation due to inhibition of ADPGKAS1 is mostly restored after overexpression of LSD2.	[11]
LSD2	AGS and MKN45 gastric cancer cell lines.	The PI3K/AKT/mTOR pathway is an important intracellular signalling pathway involved in GC prognosis. ADPGKAS1 activation induces the PI3K/AKT/mTOR signalling pathway to promote GC cell emergence and progression by regulating LSD2/KDM1B in GCs via miR3196.	[11,43]

Abbreviations: GC = gastric cancer, MGC-803 = cellosaurus cell line, SGC-7901 = human gastric cancer cell line, AGS = human gastric adenocarcinoma cell, MKN45 = human gastric adenocarcinoma, p21 = potent cyclin-dependent kinase inhibitor, BGC823 = human gastric carcinoma, GES1 = normal gastric epithelium cell, CADM4 = Cell Adhesion Molecule 4, LATS2 = large tumour Suppressor Kinase 2, KLF2 = Kruppel-like factor 2, ADPGKAS1 = long noncoding RNA ADPGK-AS1, BAP1 = BRCA1 associated protein 1, ASXL2 = ASXL transcriptional regulator 2, Caco2 = human colorectal adenocarcinoma cells, SW1116 = colon cancer cell, LoVo = colorectal cancer cell, COADREAD = colon adenocarcinoma and rectum adenocarcinoma, BLCA = bladder urothelial carcinoma, BRCA = breast invasive carcinoma, PI3K = phosphoinositide 3-kinases, AKT = protein kinase B, mTOR = mammalian target of rapamycin.

**Table 2 biomolecules-12-00462-t002:** Recent studies on LSD1/2 in liver cancer.

LSD	Model	Main Findings	Reference
LSD1	Clinical data from 188 primary HCC patients. Primary HCC cells, Male nude mice injected with tumour suspension for HCC tumour model.	LSD1 expression is positively associated with LGR5 expression and poor survival in HCC patients. Induction of LSD1 overexpression expands the pool of LGR5+ cells (cancer initiating cells), and drug resistance in HCC cells. Depletion of LSD1 attenuates the self-renewal of CSCs and their drug resistance. LSD1 reduces the H3K4me1/2 methylation at the promoters of several repressors of β-catenin signalling to enhance β-catenin activity in CSCs and enhances tumour formation	[62]
LSD1	Data from 303 HCC patients, Crispr/Cas9 for LSD1-KO SNU-423 and SNU-475 cell lines	LSD1 expression is associated with poorer outcome for overall and disease-free survival for HCC. LSD1 knockout results in reduced S-phase population and it is involved in retinoic acid (RA) pathway.	[64]
LSD1	Human HCC cell lines (Huh7, HCC-LM3, HepG2, MHCC97H and Hep3B), the normal liver cell line (LO2), and OXA-resistant liver cancer cell lines, Clinical data from 153 HCC patients, xenograft HepG2 mice models	LSD1 demethylates LINC01134 for its up-regulation and subsequently confers resistance against oxaliplatin in HCC cells. LSD1 knockdown results in deregulation of LINC01134. LSD1 expression is correlated with LINC01134 in HCC patients.	[68]
LSD1	Liver cancer Bel-7402 cells, xenograft mice models using Bel-7402 cells.	Inhibition of LSD1 leads to attenuated migration of liver cancer cells and show potential antitumor activity in vivo.	[71]
LSD1	HCC TCGA dataset, HCC cell lines HuH7, Hep3B, HepG2, SK-Hep1, PLC/PRF/5 and FOCUS.	Higher LSD1 expression is associated with poor survival in HCC. LSD1 inhibitor exhibited poor effect on HCC cell survival.	[72]
LSD1	HCC cell lines (PLC/PRF/5 and Huh7), PLC and Huh7 sorafenib-resistant cell lines, mice transfected with sh-RNA (LSD1 and LSD2) containing stable clones of HCC cell lines	LSD1 is critical for the induction of a stem-like population and inhibiting its activity attenuates stemness in sorafenib-resistant HCC cells. LSD1 inhibitors derepress the transcription of Wnt antagonists and down-regulate β -catenin signalling activity in sorafenib-resistant cells and in vivo.	[73]
LSD2	HCC tissue samples, *n* = 365.	Higher expression of LSD2 is associated with a worse prognosis.	[72]
LSD2	Sorafenib resistant Huh7 Cell line.	No change in the expression levels of LSD2. Depletion of LSD2 did not affect sensitivity to sorafenib.	[73]

Abbreviations: HCC = hepatocellular carcinoma, CSCs = cancer stem cells, siRNA = silencing RNA, OXA = oxaliplatin, LGR5 = leucine-rich repeat-containing G-protein coupled receptor 5.

**Table 3 biomolecules-12-00462-t003:** Recent studies on LSD1/2 in pancreatic cancer.

LSD	Model	Main Findings	Reference
LSD1	Gene Expression Omnibus data sets GSE15471. Human pancreatic cancer and matched paracancerous tissue samples, *n* = 28. BxPC-3 and PANC-1 human pancreatic cancer cell lines. Xenografts using BxPC-3 cells in BALB/C mice, *n* = 12.	lncRNA HOXA-AS2 promotes pancreatic cancer cell proliferation and reduces apoptosis. HOXA-AS2 binds to LSD1 in lncRNA-HOXA-AS2/EZH2/LSD1 complex to exert its oncogenic functions.	[74]
LSD1	Gene Expression Omnibus data sets GSE15471, GSE15932, GSE16515. Human pancreatic cancer tissue samples, *n* = 48. AsPC-1, BxPC-3, and PANC-1 human pancreatic cancer cell lines. Xenografts using BxPC-3 cells in BALB/c mice, *n* = 10.	lncRNA DUXAP10 promotes pancreatic cancer cells, reduces apoptosis, and is associated with poor prognosis. DUXAP10 correlates and binds to LSD1.	[75]
LSD2	Human pancreatic cancer and matched paracancerous tissue samples, *n* = 20. BxPC-3, CFPAC-1, PANC-1, SW1990 human pancreatic cancer cell lines, and HPDE6-C7 normal human pancreatic duct epithelial cell line.	LSD2 is highly expressed in pancreatic cancer. LSD2 promotes pancreatic cancer cell proliferation and reduces apoptosis. LSD2 knockdown upregulates phosphorylation of ERK1/2, Smad2, p53, cleaved PARP, cleaved Caspase-3, cleaved Caspase-7, eIF2a and Survivin.	[76]

Abbreviations: lncRNA = long noncoding RNA.

**Table 4 biomolecules-12-00462-t004:** Recent studies on LSD1/2 in colorectal cancer.

LSD	Model	Main Findings	Reference
LSD1	SW620 human CRC cell line. Xenografts using SW620 cells in BALB/c mice, *n* = 24.	LSD1 is upregulated in cells presenting cancer stem cell marker CD133. LSD1 KD impairs the stemness of CD133+ cells, decreasing cell viability, colony formation rate, migration, and invasion.	[77]
LSD1	SW620 and HT-29 human CRC cell lines.	CABYR, FOX2, TLE4 and CDH1 are 4 key LSD-1 target genes associated with proliferation, metastasis, and invasion. LSD1 downregulates CABYR expression by decreasing H3K4me1/2 and downregulates CDH1 by decreasing H3K4me2. LSD1 knockdown affects most frequently p53 pathway. LSD1 affects IG-1/mTOR pathway.	[78]
LSD1	Gene Expression Omnibus data set GSE167262. HT-29, LSD-174T, NCI-H508 human CRC cell lines. Normal human organoids derived from the ascending colon. Colon cancer organoides derived from patient-derived xenografts models 519858 162-T and 817829 284-R.	BRAF mutation is associated with poorly differentiated enteroendocrine cells. LSD1 is upregulated in early secretory cells and early enteroendocrine cells. LSD1 KD results in loss of secretory cells. LSD1 KD reduces TFF3 expression. LSD1 KD leads to loss of pS473-AKT and abrogates tumour growth and metastasis.	[79]
LSD1	CRC tissue samples, *n* =50.	Loss of LSD1 is associated with metastasis and higher TNM stages.	[80]
LSD1	CRC tissue samples, *n* = 207.	LSD1 is associated with lower TNM stages. LSD1 low expression is associated with lower disease-specific and disease-free survival.	[81]
LSD1	HT-29, SW480, HCT116, LoVo, and RKO human CRC cell lines. AGS human gastric cancer cell line.	LSD1 is upregulated in PIK3CA mutant CRC compared to PIK3CA wt CRC. LSD1 increases pS473-AKT by scaffolding the CoREST complex. LSD1 enhances EMT-associated gene programmes in PIK3CA mutant cells. LSD1 regulates protein stability of Snail by regulating AKT. LSD1 is required for EGF induced migration mediated by AKT-GSK3β-Snail pathway.	[82]
LSD1	HCT116 human CRC cell line.	RIOK1 promotes CRC cell proliferation and migration through PI3K/AKT pathway. LSD1 demethylates RIOK1 stabilizing it.	[83]
LSD1	HCT116 and SW48 human CRC cell lines. HEK293T human embryonic kidney cell line.	USP38 binds and deubiquitinase LSD1, enhancing the activity of the signalling pathways activated by LSD1.	[84]
LSD1	Tissue microarray containing 100 human CRC cases. HT-29 and HCT116 human CRC cell lines.	TNC expression is associated with poor clinical outcomes, proliferation, and migration. TNC is positively correlated with the LSD1 protein in CRC. TNC KD decreases LSD1 expression.	[85]
LSD1	SW620, SW480, DLD-1, HTC116, and HT-29 human CRC cell lines. NCM460 normal human colon epithelial cell line. HEK293T human embryonic kidney cell line.	LSD1 and TSPAN8 are overexpressed in CRC. LSD1 upregulates TSPAN8 expression by reducing H3K9me2 occupancy on TSPAN8 promoter. TSPAN8 enhances tumorigenicity and EMT in CRC cells in an LSD-1 dependent manner. LSD1 and TSPAN8 KD results in an upregulation of E-cadherin and ZO-1, and a downregulation of n-cadherin, Vimentin, Slug and Snail.	[86]
LSD1	HCT116 human CRC cell line. HeLa human cervical cancer cell line.	LSD1 inhibition modifies the lipidome of cancer cells, specially it dysregulates sphingolipids and glycolipids. LSD1 inhibition enhances bioactive lipids such as ceramides and sphingomyelin which are involved in signalling pathways such as apoptosis.	[87]
LSD1	CRC tissue samples, *n* = 58. DLD-1, HCT116, SW480, and SW620 human CRC cell lines.	lncRNA DUXAP10 is positively associated with CRC cell proliferation, tumour size, advanced TNM stages and lymph node metastasis. LSD1 interacts with DUXAP10 and decreases p21 and PTEN expression.	[88]
LSD1	Human CRC and matched paracancerous tissue samples. DLD-1 and SW480 human colorectal cancer cell lines.	lncRNA DUXAP8 is positively associated with CRC cell proliferation, tumour size, and advanced TNM stages. LSD1 interacts with DUXAP8 and promotes CRC cell proliferation.	[89]
LSD1	Human CRC and matched paracancerous tissue samples, *n* = 60. HT-29, HCT116, SW480 and DLD-1 human colorectal cancer tissue samples. NCM460 human colon epithelial normal cells.	lncRNA ZEB2-AS1 is associated with increased proliferation rate, tumour size, higher TNM stage and lymph node metastasis. LSD1 interacts with ZEB-AS1 promoting cell proliferation.	[90]
LSD1	Human CRC and matched paracancerous tissue samples, *n* = 69. HCT116, DLD-1, SW480, SW620, HT-29, and LoVo human CRC cell lines.	lncRNA HOXA-AS2 is associated with increased proliferation rate, tumour size, higher TNM stage and lymph node metastasis. LSD1 interacts with HOXA-AS2 silencing p21 and KLF2 transcription and promotes cell proliferation.	[91]
LSD1	Human CRC and matched paracancerous tissue samples, *n* = 98. LoVo and HCT116 human CRC cell lines.	miR-137-3p is negatively associated with the invasiveness of CRC cells. LSD1 is regulated by miR-137-3p and is involved in CRC cell proliferation, adhesion, and invasion. LSD1 and miR-137-3p respond to hypoxia.	[92]
LSD2	LoVo, HCT116, SW1116, andCaco2 human CRC cell lines. Xenografts assays using LoVo cells in BALB/c mice, *n* = 6.	LSD2 is upregulated in CRC. LSD2 reduces Bcl-2 and increases cleaved caspase 3, cleaved caspase 9 and BAX levels. LSD2 downregulates p53 expression and p21 and drives the cell cycle through p53-p21-Rb.	[44]

Abbreviations: CRC = colorectal cancer, KD = knockdown, wt = Wildtype, EMT = epithelial-mesenchymal transition, TNC = tenascin-C, lncRNA = long noncoding RNA.

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
