# Peer review of "The Role of LSD1 and LSD2 in Cancers of the Gastrointestinal System: An Update"

_biomolecules, 2022, doi:10.3390/biom12030462_

Round 1

Reviewer 1 Report

The reviewer thought that the authors sincerely revised the paper according to the reviewer’s comments and questions. The reviewer has however one comment to recommend fix a spelling mistake for present revised version of this article.

#In figure 2, there was still remaining a spelling mistake “Oxalaplatin.” Please correct this to “Oxaliplatin.”

Author Response

Reviewer 1

The reviewer thought that the authors sincerely revised the paper according to the reviewer’s comments and questions. The reviewer has however one comment to recommend fix a spelling mistake for present revised version of this article.

#In figure 2, there was still remaining a spelling mistake “Oxalaplatin.” Please correct this to “Oxaliplatin.”

Response

We appreciate the acknowledgement of our efforts in revising the manuscript and the spelling mistake has been rectified in the final revised manuscript.

Reviewer 2 Report

Manuscript entitled "The role of LSD1 and LSD2 in cancers of the gastrointestinal system: an update"

This work is well-written and could be acceptable pending minor modifications as followings:

  1. A more systemic and updating review of LSD1 and LSD2 should be added into introduction.
  2. The therapeutic implications of LSD1 and LSD2 in various cancer types should be addressed. 

Author Response

Reviewer 2

This work is well-written and could be acceptable pending minor modifications as followings:

  1. A more systemic and updating review of LSD1 and LSD2 should be added into introduction.
  2. The therapeutic implications of LSD1 and LSD2 in various cancer types should be addressed. 

Response

  1. The introduction has been updated according to the suggestion from the reviewer.
  2. Another section titled ‘Therapeutic implications of LSD1 and LSD2 in various cancers’ has been added in the manuscript (before the Conclusions) according to the suggestion from the reviewer.

This manuscript is a resubmission of an earlier submission. The following is a list of the peer review reports and author responses from that submission.

Round 1

Reviewer 1 Report

I judged this review article is not suitable for publication. The reasons are as follows.

  1. LSD1 and LSD2 do not have a jumonji domain and are not categorized into jumonji domain-containing proteins. Therefore, this review article does not seem to be suitable to this special issue.
  1. The introduction is verbosely written and not summarized well. It will make readers painful.
  1. Although I don’t check all the references, the reference works are slovenly.

- Page numbers and volumes of the several references are missing.

-Some mentions are not suitable to the corresponding references (e.g. ref. 57., 101, etc.).

- ref. 72-81 are missing in the main text.

  1. Figures are meaningless.

-Figure 1: the two panels are same.

-Figures 1 and 2: “Acetylation” is hardly mentioned

-Figures 1-4: the relationships in each factor are unclear.

  1. This article is a mere juxtaposition of the information reported previously. There are little author’s opinions or considerations throughout the paper and I do not think that this paper inspires journal readers.

Reviewer 2 Report

Posttranslational modifications (PTMs) including methylation contribute to various types of carcinogenesis, tumor progression and metastasis/recurrence. The reviewer is also interested in the function for malignancy by methylation and demethylation. In this study, the authors summarized and reviewed about LSD1 (Lysine specific demethylase 1) and LSD2 (Lysine specific demethylase 2). LSD1 has been reported as the first discovered demethylase in 2004. Both LSD1 and LSD2 also have the crucial role for tumor progression, moreover, the development of molecular-targeted therapies focused these two enzymes is also undergoing through translational research. The reviewer is also interested in this review, there are however some questions to clear up more as below:

#1: Focused and featured topics in this report were the roles of LSD1 and LSD2 in cancers such as gastric cancer, liver cancer, pancreatic cancer and colorectal cancer. Generally speaking, “gastrointestinal tract” does not include in liver and pancreas. Please re-think and change the title.

#2: In figure 2, “Oxaliplatin” was one of the affecting factors by LSD1 in figure 2, however, oxaliplatin is not generally a key drug for liver cancer at present. This figure may be misleading to the readers and recommend to be fixed.

#3: LSD1 and LSD2 also have the demethylase activity for not only histone but also non-histone protein and these actions are thought to be important for tumor progression. Please mention more about LSD1function for non-histone protein and add reference below as examples.

  1. p53 is regulated by the lysine demethylase LSD1.

Huang J et al. Nature. 2007; 449(7158): 105-8.

  1. Demethylation of RB regulator MYPT1 by histone demethylase LSD1 promotes cell cycle progression in cancer cells.

Cho HS et al. Cancer Res. 2011; 71(3): 655-60.

  1. LSD1 demethylase and the methyl-binding protein PHF20L1 prevent SET7 methyltransferase-dependent proteolysis of the stem-cell protein SOX2.

Zhang C et al. J Biol Chem. 2018; 293(10): 3663-3674.

#4: Please add this below report into table 4 and references because a large number of hepatocellular carcinoma (HCC) patients were entried and subsequently functional analyses were performed in detail.

  1. Deregulation of the Histone Lysine-Specific Demethylase 1 Is Involved in Human Hepatocellular Carcinoma.

Kim S et al. Biomolecules. 2019; 9(12): 810.

#5: The reviewer thinks and believes that LSD1 and LSD2 are potential candidates for molecular-targeting drugs in near future. Please discuss more about current status and prospects of LSD1/2 inhibitors in “Discussion” section.

  1. LSD1/KDM1A inhibitors in clinical trials: advances and prospects.

Fang Y et al. J Hematol Oncol. 2019; 12(1): 129.